# Pilot Study: Quantitative Photoacoustic Evaluation of Peripheral Vascular Dynamics Induced by Carfilzomib In Vivo

**DOI:** 10.3390/s21030836

**Published:** 2021-01-27

**Authors:** Thi Thao Mai, Manh-Cuong Vo, Tan-Huy Chu, Jin Young Kim, Chulhong Kim, Je-Jung Lee, Sung-Hoon Jung, Changho Lee

**Affiliations:** 1Department of Artificial Intelligence Convergence, Chonnam National University, 77 Yongbong-ro, Buk-gu, Gwangju 61186, Korea; 196286@jnu.ac.kr; 2Research Center for Cancer Immunotherapy, Chonnam National University Hwasun Hospital, 264, Seoyang-ro, Hwasun-eup, Hwasun-gun, Jeollanam-do 58128, Korea; cuong44cnsh@yahoo.com (M.-C.V.); huychutan2010@gmail.com (T.-H.C.); drjejung@chonnam.ac.kr (J.-J.L.); 3Department of Creative IT Engineering and Electrical Engineering, Pohang University of Science and Technology (POSTECH), 77 Cheongam-ro, Nam-gu, Pohang, Gyeongbuk-do 37673, Korea; ronsan@postech.ac.kr (J.Y.K.); chulhong@postech.edu (C.K.); 4Department of Hematology-Oncology, Chonnam National University Hwasun Hospital, Hwasun, Jeollanam-do 58128, Korea; 5Department of Nuclear Medicine, Chonnam National University Medical School & Hwasun Hospital, Hwasun, Jeollanam-do 58128, Korea

**Keywords:** carfilzomib, peripheral vasculature, photoacoustic microscopy, quantitative analysis

## Abstract

Carfilzomib is mainly used to treat multiple myeloma. Several side effects have been reported in patients treated with carfilzomib, especially those associated with cardiovascular events, such as hypertension, congestive heart failure, and coronary artery disease. However, the side effects, especially the manifestation of cardiovascular events through capillaries, have not been fully investigated. Here, we performed a pilot experiment to monitor peripheral vascular dynamics in a mouse ear under the effects of carfilzomib using a quantitative photoacoustic vascular evaluation method. Before and after injecting the carfilzomib, bortezomib, and PBS solutions, we acquired high-resolution three-dimensional PAM data of the peripheral vasculature of the mouse ear during each experiment for 10 h. Then, the PAM maximum amplitude projection (MAP) images and five quantitative vascular parameters, i.e., photoacoustic (PA) signal, diameter, density, length fraction, and fractal dimension, were estimated. Quantitative results showed that carfilzomib induces a strong effect on the peripheral vascular system through a significant increase in all vascular parameters up to 50%, especially during the first 30 min after injection. Meanwhile, bortezomib and PBS do not have much impact on the peripheral vascular system. This pilot study verified PAM as a comprehensive method to investigate peripheral vasculature, along with the effects of carfilzomib. Therefore, we expect that PAM may be useful to predict cardiovascular events caused by carfilzomib.

## 1. Introduction

Carfilzomib is a second-generation proteasome inhibitor, mainly used to treat multiple myeloma (MM) [1]. Combined with dexamethasone or lenalidomide and dexamethasone, carfilzomib has proven to significantly improve the survival outcomes of relapsed refractory MM patients in randomized phase 3 clinical trials [2]. Although carfilzomib is generally well tolerated, it is associated with adverse cardiovascular events, including hypertension, congestive heart failure, and coronary artery disease. In a small prospective study, patients treated with carfilzomib exhibited more cardiovascular events than those treated with bortezomib (51% vs. 17%, *P* = 0.002), and patients who experienced cardiovascular events exhibited significantly inferior progression-free survival (*P* = 0.01) and overall survival (*P* < 0.001) [3]. While the mechanisms of cardiovascular adverse events have not been completely elucidated, one study suggests that endothelial dysfunction in the coronary vasculature caused by proteasome inhibition impacted the development of cardiovascular adverse events [4]. Generally, endothelial dysfunction is not limited to coronary vasculature and may affect peripheral vasculature, which could predict cardiovascular events [5]. Therefore, we hypothesize that proteasome inhibitors change the peripheral vasculature, and the identification of these changes may be a useful and convenient method to predict cardiovascular events using carfilzomib.

Various imaging modalities already contribute to the investigation and monitoring of the peripheral vasculature and its changes. Computed tomography (CT) and magnetic resonance imaging (MRI) can visualize the peripheral vasculature by injecting contrast agents [6,7]. Unfortunately, these are limited due to their relatively low resolution, radiation exposure, exogenous contrast, large system size, high post-processing time, and high cost. Ultrasound imaging (USI), with the Doppler effect, is also utilized to obtain vasculature information. However, Doppler USI is limited to the visualization of microvasculatures. Recently, USI has been used to visualize the microvasculatures of a mouse brain using microbubble-contrast agents and fast GPU processing of big data [8]. Optical imaging techniques, such as confocal microscopy (CM), multiphoton microscopy (MPM), and optical coherence tomography (OCT), have been utilized to visualize microvessels, based on a focused beam with a spatial resolution less than 10 μm [9,10,11,12,13,14]. Nevertheless, due to light scattering, CM and MPM are limited by a penetration depth in the range of hundreds of micrometers. Furthermore, these methods require additional contrast agents. Although OCT enables a better imaging depth (approximately 1 mm), it still requires complex signal processing. 

One of the most notable imaging methods spotlighted in recent times is photoacoustic microscopy (PAM). Based on the photoacoustic (PA) effect that generates ultrasonic waves from light absorption, it inherits the hybrid imaging property of optical imaging and USI [15,16,17]. Owing to less scattering of ultrasound in the tissue, PAM can achieve a penetration depth of up to several centimeters and completely break the optical transport mean free path (i.e., ~1 mm) of pure optical imaging techniques. Depending on the crucial imaging setup, PAM can also provide multi-scale resolution from nano- to micro-scales by maintaining its optical absorption contrast. These benefits overcome the limited resolution of USI [18,19,20]. Moreover, PAM is a compact, inexpensive, non-ionized, label-free, functional, and real-time imaging modality [21,22,23,24,25]. It particularly aids the visualization of the distribution of intrinsic molecules in the body, such as hemoglobin, lipid, melanin, and proteins [26,27,28,29,30,31]. 

Additionally, PAM is widely utilized to visualize vasculatures and monitor their changes. For peripheral vascular visualization, B. Rao et al. reported an optical resolution-PAM (OR-PAM) system to visualize mouse ear microvasculatures [32]. A dual-modality imaging system combining PAM with USI was reported by Y. Tang [33] to achieve anatomical and functional information of a mouse’s hind paw. Moreover, aiming towards human implementation, a PAM-based system for human peripheral arteries was developed [34]. For drug monitoring, Y. Liu et al. used the PAM system to assess norepinephrine via cerebral vessels [35]. Additionally, R. Bi et al. investigated orthotopic glioma blood vessels under the effect of combretastatin A4 phosphate [25]. L. Nie et al. proved PAM’s effectiveness in monitoring nanocarrier-enhanced chemotherapy response in the early stage of brain tumor treatment [36]. 

However, most of these peripheral vascular visualization studies are interpreted qualitatively. To improve the accuracy in assessing the abnormal vascular, vessel quantitative parameters need to be extracted such as diameter, density, length fraction, and fractal dimension. One of the most common parameters of a blood vessel, the diameter, represents its size. A change in temperature causes a change in the vascular diameter [37]. The phases of perfusion of NaCl are also reflected through variation in brain vessel diameter [38]. Furthermore, a change in retinal vessel diameter has been shown to be a sign of stroke [39,40,41]. The density and length fraction of vessels implies the value of the whole vessel area and the total vessel length, respectively. These parameters play an important role in evaluating blood vessel abnormalities of the retina such as hyperoxia or glaucoma [42,43,44,45]. In addition, the vascular density was monitored for malignant and non-malignant skin [46,47]. Fractal dimension describes how completely a vascular network fills a space [48]. On the other hand, it indicates the vessel tortuosity and branching [49]. Hence, fractal dimension is used to describe the complexity of biological structures, including the coronary [50], parafoveal capillary network [51] and tumor vascular network [48]. Therefore, these parameters reveal the characteristic features of a blood vessel, which are especially useful in monitoring abnormalities that occur in the blood vessels.

In this study, we focus on photoacoustically monitoring the dynamics of peripheral vasculatures in mouse ears under the effect of carfilzomib for 10 h using high-resolution label-free OR-PAM. The same follow-up imaging process was repeated with the injection of bortezomib and PBS. After acquiring three-dimensional OR-PAM data for all the cases, the OR-PAM MAP images were reconstructed and analyzed. For a comprehensive assessment, the peripheral vasculature’s structural morphology was also monitored, and the quantitative vasculature evaluation process, including diameter, density, length fraction, fractal dimension, and PA signal, was implemented.

## 2. Materials and Methods

### 2.1. Carfilzomib Solution

The proteasome inhibitors carfilzomib (Kyprolis; Onyx Pharmaceuticals, San Francisco, CA, USA) and bortezomib (Velcade; Millennium Pharmaceuticals, Cambridge, MA, USA) were dissolved in sterile, 0.9% (*v*/*v*) normal saline immediately before use. Carfilzomib and bortezomib, at a dose of 10 and 0.5 mg/kg in 100 µL PBS, respectively, were administrated to the mice by intravenous injection. The doses of carfilzomib and bortezomib were determined based on previous studies [52]. In clinical practice, the bortezomib dose is constant. However, the dose of carfilzomib depends on the combination of drugs or the administration schedule. We selected a higher dose of carfilzomib in this experiment, as cardiovascular events are associated with higher carfilzomib doses [53].

### 2.2. Animal Preparing 

All experimental animal procedures followed laboratory animal protocols approved by the Institutional Animal care and use committee of Chonnam National University Hwasun Hospital (CNU IACUC-H-2018-68). Healthy six-to-eight-week-old female BALB/c (H-2d) mice, weighing ~20 g, were purchased from Orient Bio (Iksan, Korea) and maintained under specific pathogen-free conditions. Each mouse was anesthetized with an intraperitoneal injection of Ketamine (80 mg/kg)/Xylazine (12 mg/kg). After removing the downy hairs on its ear, the mouse was placed on a homemade animal holder. An isoflurane system (Luna Vaporiser, NorVap international LTD, Barrowford, UK) for gaseous anesthetization and a temperature maintain bed were used in long-term in vivo observation (10 h) to maintain stable body conditions of the mouse. The energy of the illuminated laser pulse on the mouse skin was approximately 5 mJ/cm^2^ below the American National Standards Institute (ANSI) safety limit (20 mJ/cm^2^).

### 2.3. Optical-Resolution Photoacoustic Microscopy

Figure 1 describes the schematic of the OR-PAM system. Trigger signals from the data acquisition (DAQ) board (PCIe-6321, NI instruments, Austin, TX, USA) were sent to a diode laser (SPOT-10-200-532, Elforlight, Daventry, UK) to operate the primary laser beam at a wavelength of 532 nm. The fired laser beam, with a duration of 6 ns, was spatially filtered by an iris (SM1D12, Thorlabs, Newton, NJ, USA). The reshaped laser beam was reflected between a pair of mirrors to transfer it to a collimator (F280APC-A, Thorlabs, NJ, USA; f = 18.07 mm, NA = 0.15). After coupling into a single-mode optical fiber (P1-405BPM-FC-1, Thorlabs, NJ, USA), the laser beam was efficiently delivered to the second collimator (F260APC-A, Thorlabs, NJ, USA, f = 15.01 mm, NA = 0.17). An objective lens (AC254-060-A, Thorlabs, NJ, USA), with a focal length of 60 mm, was utilized to focus the collimated laser beam. A high-precision zoom housing (SM1ZM, Thorlabs, NJ, USA), which could adjust the optical focal plane, with a maximum of 4.1 mm along the *z*-axis, was used to mount the objective lens. The focused beam was passed through a correction lens and made to penetrate a hand-made beam combiner. Inside the combiner, the beam was reflected by a layer created by a normal and an aluminum-coated prism. Before the focused beam illuminated the sample, the focused beam was redirected by reflecting it on the mirror of the MEMS scanner (OptichoMS-001, Opticho Inc., Ltd., Pohang, Korea) and passed through a tank. The confocal and co-axial alignment of the incident laser beam and ultrasound occurred between the combiner and the MEMS scanner’s mirror. The omnidirectional photoacoustic wave was generated immediately after the illumination of the sample. With the focused support of a concave lens located on the right side of the combiner, the PA wave easily passed through the combiner and was detected by a high-frequency ultrasonic transducer (V214-BC-RM, 50 MHz, Olympus, Tokyo, Japan). The signal acquisition part aimed to improve and convert the PA waves into image information. Two RF-amplifiers (ZX60-3018G-S+, Mini-Circuit, Brooklyn, NY, USA) were used to amplify the released PA wave from the sample. Then, a high-speed digitizer (ATS9371, AlazarTech, Pointe-Claire, QC, Canada) digitalized the signal to convert it into primary image information. A linear stepper motor stage (L-509-10SD00, Physik Instrumente, Karlsruhe, Germany) on the *y*-axis was associated with the MEMS scanner on the *x*-axis to obtain the 3D image data. This integrated scan system was used to drive the beam to scan the object’s surface, with a speed of 25 Hz B-scan. The data acquisition time for a mouse ear with an area of 10 × 12 mm is 180 s. The measured lateral and axial resolutions were 12 and 45 µm, respectively [21]. A LabVIEW program (National Instruments, Austin, TX, USA) was used to operate the OR-PAM system. The released image information was reconstructed and analyzed with MATLAB (R2016a, Mathworks, Natick, MA, USA).

### 2.4. Quantification Evaluation Process of OR-PAM Image 

Figure 2 illustrates the quantitative evaluation process of the OR-PAM images and presents the two main steps involved: (1) image segmentation and (2) quantitative parameter extraction. In the segmentation stage, the multi-scale Hessian filter, intended specifically for vascular objects, was used. The Hessian filter could separate the boundaries between blood vessels and the background based on the second-order gradient of the image [54]. By utilizing multiple scales, multi-size vessels were segmented. The binary image was obtained using the adaptive threshold method [55]. Finally, morphological operations were used to obtain a skeleton map. A skeleton map showing only the centerline of the object was also obtained. These two maps were used as input sources for the next step, i.e., extraction. The four required parameters were extracted by implementing some methods on the binary and skeleton maps. Table 1 summarizes the formulas of the parameters.

The shortest distance calculated by Euclidean transform for a certain vertical section of a vessel was assumed as the diameter of that section. The diameter of a single vessel was measured as the average distance transform along the skeleton of that vessel. Thus, only the vessel area will contribute to the average diameter vessel of an image. Equation (1) indicates the average diameter vessel for an image, where the Euclidean distance transform was implemented on the corresponding binary map. In Equations (2) and (3), the percentage area covered by the vessel was used to define the density and length fraction parameters. Length fraction only considers the existence of the vessel by counting the number of white pixels on the skeleton map to figure out the ratio of white pixels and total pixels. Thus, dilation or constriction of the vessel does not impact length fraction. In contrast, density was influenced by both the diameter and length fraction of the vessel. To compute density, we counted the number of white pixels on the binary image and divided it by the number of the total pixels of the image. The fractal dimension was calculated by the box-counting method [56]. The binary map was divided into square boxes of the same size, called the unit boxes. The number of unit boxes needed to cover a vessel was counted. The logarithm of the unit box size with the corresponding number of boxes produces a curve; the fractal dimension is determined as the absolute value of that curve (Equation (4)). The PA signal was also monitored to comprehensively evaluate the change in blood vessels comprehensively. PA signal describes the average incident photon energy that was absorbed by blood vessels. For all formula, m and n are sizes of the image (MAP image, binary map and skeleton map share same size); *i* and *j* are the calculated coordinates of the pixel. Total pixels *T*(*i*, *j*) = *m* × *n*.

## 3. Results

### 3.1. In Vivo OR-PAM Observation for the Peripheral Vasculatures after Carfilzomib Solution Injection

The mouse ear’s peripheral vasculature (Figure 3k) was monitored for 10 h before and after injecting the carfilzomib solution. Figure 3a–j shows the acquired OR-PAM MAP images with the image color scale fixed on all images to investigate signal change monitoring. Figure 3a is a control image, showing that the main vasculatures and microvessels are generally well distributed. The OR-PAM MAP image presented in Figure 3b was obtained 10 min after carfilzomib solution injection. The bright color of the vessels indicates a significant increase in the PA signal intensity in these vessels. Several bleeding spots were formed, especially along the large blood vessels, and several new microvasculatures, that had appeared suddenly, had formed. The effect of the carfilzomib seemed to have peaked at 30 min, based on the image presented in Figure 3c. Although the PA signal in the major vasculatures had only marginally changed compared to that at the previous imaging timepoint, the areas where the microvasculatures were present continued to expand. Figure 3d–j presents the OR-PAM MAP images, obtained 1, 2, 3, 4, 6, 8, and 10 h after the injection, respectively. During this time period, the PA signal decreased gradually. Specifically, at the 4 h mark, the microvessels had begun to disappear slowly and completely disappeared at the 10 h mark. The PA signal gradually recovered to a steady state, similar to that shown in the control image.

For more detailed analysis, we randomly monitored nine small regions representing large vasculatures (ROI 1 and ROI 8) and microvasculatures (ROI 2, ROI 3, ROI 4, ROI 5, ROI 6, ROI 7, and ROI 9), as shown in Appendix A. For the large vasculatures, ROI 8, on the margins of the mouse ear, showed the least amount of change during the follow-up imaging. In contrast, ROI 1, which included bleeding points, exhibited an increase in the signal approximately 30 min after the injection and a gradual decrease thereafter. However, this increase was only approximately 50% to 100% compared to the initial observation value. Overall, the variation in the signal due to carfilzomib’s effect was observed in all areas of the small blood vessels. Within the first 30 min, it was observed that the closer the ROI was to the center, the larger the extent of changes. For example, ROI 2, ROI 7, ROI 6, and ROI 4 showed changes in the PA signal ranging between 150% and 400%, and for the other parameters, only 50–120% changes were present. Meanwhile, the ROIs in the far center (ROI 3 and ROI 5) exhibited an increase in the PA signal by approximately 50–100%, and only 10–50% changes were exhibited for the other parameters. Located at the edge of the ear, where there were very few blood vessels, ROI 9 exhibited changes that were distinct from those of the rest of the ROIs. However, all ROIs began to stabilize at the 4-h mark.

### 3.2. In Vivo OR-PAM Observation for the Peripheral Vasculatures after Bortezomib Solution Injection

The experiment was repeated by substituting carfilzomib with bortezomib. The OR-PAM MAP image of the control presented in Figure 4a, obtained before injecting the bortezomib solution, clearly shows all the vasculatures not shown in Figure 4k. Unlike carfilzomib, bortezomib did not significantly increase the vessel’s signal after the first 10 min of the injection (Figure 4b). The mouse ear was monitored for approximately 10 h with the OR-PAM MAP images. Figure 4c–j depicts the mouse ear 30 min and 1, 2, 3, 4, 6, 8, and 10 h after the bortezomib injection. The same color was observed in the OR-PAM MAP images with the big vasculatures, indicating no increase in the PA signal. Moreover, no bleeding area was detected, and no small vessel appeared. This indicates that steady state was maintained for 6 h after injection. In Appendix A, the MAP images and quantification results for nine small ROIs are shown, in which ROI 1, ROI 2, ROI 3, ROI 4, and ROI 9 indicate small vessel areas, and ROI 5, ROI 6, ROI 7, and ROI 8 describe large vessel areas. The stability, monitored using MAP images, maintained during 6 h post-injection, was observed at all the ROIs. Furthermore, variation in the quantitative values of all the parameters was only approximately 20% compared to those at the control time. Peak value at the 8 h mark was only noticeable at ROI 3, ROI 8, and ROI 9, which were near the mouse ear’s edges. However, the increase was only approximately 50–60%, which decreased after 10 h of monitoring.

### 3.3. In Vivo OR-PAM Observation for the Peripheral Vasculatures of after PBS Injection

An experiment with PBS injection is required to standardize and accurately and objectively evaluate the effectiveness of carfilzomib and bortezomib. Figure 5 displays the OR-PAM MAP images of the monitoring process. Figure 5a shows the condition before the injection, Figure 5b–j shows the observations from the 10 min to the 10 h mark. Figure 5k shows the photograph of the mouse’s ear. Under stable mouse conditions, the PBS did not seem to affect the stability of the large blood vessels and small capillaries before and after the injection. As depicted in Appendix A, for OR-PAM MAP images and graphs of quantification results, there is no general trend in the change in signals at all the ROIs. The highest variations, smaller than 100%, belong to ROI 3 and ROI 5 for the length fraction and PA signal, respectively. For the remaining ROIs, the values were always less than 50% for all parameters.

### 3.4. Quantitative Evaluation of OR-PAM Data

A quantitative assessment is a necessary supplement, as only comparing images is insufficient to accurately assess the effects of carfilzomib and bortezomib on the peripheral vasculature of the mouse ear. The quantitative process presented in Section 2.4, for the carfilzomib, bortezomib, and PBS injection cases, was applied. After obtaining five parameters for each case, the values of all three cases were compared for each parameter: (1) PA signal, (2) diameter, (3) density, (4) length fraction, and (5) fractal dimension. The values were presented as a percentage difference from the control value to unit normalize all the parameters. Figure 6 shows the results of this process. For all parameters, the variation in the signal, caused by carfilzomib, always gives the maximum value, 45% (PA signal at the 10-min timepoint) or 48% (density at 3-h timepoint). Meanwhile, the values corresponding to bortezomib and PBS were almost lower than 20%. The PA signal comparison shown in Figure 6a highlights that the value corresponding to carfilzomib tends to increase rapidly 10 min after the injection and then gradually decrease for 10 h. The mean value with carfilzomib is found to be 18.2%, almost eight times higher than that observed with bortezomib (2.3%) and 5.5 times higher than that observed with PBS (3.3%). This proves that image-based judgment is plausible. The graphs comparing diameter (Figure 6b), density (Figure 6c), length fraction (Figure 6d), and fractal dimension (Figure 6e) also show a similar trend. With the exception of the 3-h mark and 6-h mark, the values observed in the bortezomib case did not change significantly. In the PBS case, the values were stable at all timepoints. Specifically, at the 30-min timepoint, the higher values observed in the carfilzomib case compared to those of the bortezomib and PBS cases were 51.7% and 54.2% for the density value, 30.7% and 36.7% for the length fraction value, and 19.7% and 16.8% for the diameter, respectively. The effect of carfilzomib is evident in Figure 6e, depicting the fractal dimension, which is approximately 10 times higher than that observed in the bortezomib and PBS case.

## 4. Discussion

We successfully monitored the effects of carfilzomib on the peripheral vasculature for over 10 h using a hybrid process of OR-PAM imaging and quantitative evaluation. The results show the mouse ear’s peripheral vasculature dynamic in morphological and quantitative evaluations. Among the morphological aspects, the vascular changes were observed 10 min after injecting the carfilzomib solution, with the appearance of bleeding spots along with vessels. The emergence and strong spread of microvasculatures between 10 and 30 min indicate the occurrence of the peak effect of carfilzomib during the first 30 min after the injection. It was observed that the expansion of the microvasculature area slowly decreased, and the mouse ear vascular network recovered to the steady state after 10 h. The quantitative evaluation was used to evaluate five blood vessel parameters, including the PA signal, diameter, density, length fraction, and fractal dimension. A vascular abnormality could be identified based on the changes in all the vessel properties, which were closely related. The diameter and length fraction present the size and the perfusion of the vessel, respectively. The density value, influenced by the diameter and length fraction, gives an overview of the general situation of an area. The fractal dimension indicates the complexity of the vascular network. Microvasculatures are typically more complex than large vasculatures [57]. The results of this study show that the diameter, density, length fraction, and fractal dimension follow a similar trend. These parameters significantly increased and peaked 30 min after the injection. This seems to indicate an increase in blood pressure resulting from the effect of carfilzomib. When the blood vessels dilated, the perfusion increased and expanded the area of the blood vessels simultaneously. Additionally, an increase in the length fraction with the fractal dimension implies the appearance of new capillaries. Beginning from the 1 h mark, values of all the parameters started to decrease slightly and then maintained a stable value until the 10 h mark. All these changes were relatively consistent with the morphological visualization captured in the OR-PAM MAP images. Unlike other parameters, the PA signal is not a vasculature morphological property as it is reflected by the intensity of the photons absorbed by the blood. The value of the PA signal exhibited a significant increase in the first 10 min, after which it rapidly decreased. At a wavelength of 532 nm, the strong light absorption of the blood made our system highly sensitive to changes in the blood. Hence, the maximum value change and the post-peak reduction rate of the PA signal were always the highest.

In addition to carfilzomib, we conducted the same experiment using bortezomib and PBS. The average values calculated for all the parameters using carfilzomib were eight times higher than those observed with bortezomib and 5.5 times higher than those with PBS, emphasizing that carfilzomib has a stronger effect on the peripheral vasculature than bortezomib and PBS. However, especially at the 30-min mark, the higher values in the values observed with carfilzomib with respect to those with bortezomib and PBS were 51.7% and 54.2% for the density value, 30.7% and 36.7% for the length fraction value, and 19.7% and 16.8% for the diameter, respectively. As mentioned in Section 2.4, the change in density depends on the changes in diameter and length fraction. Thus, the contribution to the density value in this experiment was mainly from the length fraction, indicating that the length fraction changes the most under the effect of carfilzomib. It also reflects that carfilzomib has a stronger impact on the capillaries than large blood vasculatures, because diameter depends on large blood vessels, while length is determined by small blood vessels. 

In myeloma cells, the anti-myeloma effect induced by the proteasome inhibitor results from the accumulation of regulatory proteins within the endoplasmic reticulum, which further induces the apoptosis cascade [58]. Likewise, proteasome inhibition in myocardial cells leads to an abnormal accumulation of ubiquitinated proteins and can result in cardiac damage and heart failure [59]. Additionally, endothelial dysfunction caused by proteasome inhibition in the vasculature is associated with cardiovascular adverse events, such as hypertension and myocardial ischemia. Proteasome inhibition affects signaling in vascular smooth muscle endothelium and leads to increased vascular tone and coronary resistance [4]. Therefore, all proteasome inhibitors can theoretically cause cardiovascular adverse events. However, these effects are different when using bortezomib and carfilzomib, owing to differences in their pharmacodynamics and pharmacokinetic characteristics. The effect of carfilzomib is irreversible, and it is a more selective inhibitor for the β5 domain, with chymotrypsin-like activity of the 20s subunit of the proteasome, compared to bortezomib. Efentakis P et al. [60] demonstrated that bortezomib did not affect cardiac function. However, carfilzomib caused deterioration of the left ventricular function through increased PP2A activity and inhibition of the AMPKα pathway in the in vivo mouse model. In clinical studies, carfilzomib cases had a higher incidence of cardiovascular adverse events than bortezomib cases [3,61,62]. However, the effect of carfilzomib on the peripheral vasculature has not been reported.

Despite the success in monitoring the effects of carfilzomib on the peripheral vasculature with OR-PAM, our approach has certain limitations. First, the limit of the low penetration depth of its system (~1 mm) is the main hindrance to OR-PAM to become a clinical imaging method even it has super high resolution. Hence, for human applications, a proper handheld probe and the near-infrared laser should be implemented [63,64,65]. Second, although it is able to detect the concentration of hemoglobin, the use of single wavelength laser (i.e., 532 nm) is limited to provide further functional information including oxygenated (HbO_2_) and deoxygenated (HbR) hemoglobin. Thus, our system cannot directly provide the functional information of vessel such as oxygen saturation (SO_2_) and cerebral metabolic rate of oxygen (CMRO_2_) in the peripheral vasculature. By combining different wavelength lasers, it becomes able to offer metabolism information directly [66].

## 5. Conclusions

In conclusion, we implemented PAM for the mouse ear peripheral vasculature visualization within 10 h after carfilzomib, bortezomib, and PBS injection. Not only morphological, but also vascular parameters such as PA signal, diameter, density, length fraction, and fractal dimension, were successfully evaluated. Carfilzomib induces a strong effect on the peripheral vascular system during the first 30 min after injection, which can be which can be qualitatively visualized by the appearance of bleeding spots and capillary in the MAP images. Moreover, for the quantitative results, all vascular parameters significantly increase up to 50%. In contrast, bortezomib and PBS do not have much impact on the peripheral vascular system. As a pilot study, we monitored the effects of carfilzomib on the peripheral vascular system with the PAM technique and its quantitative analysis. Therefore, we expect PAM to be able to use an important tool to predict cardiovascular events triggered by carfilzomib.

## Figures and Tables

**Figure 1 sensors-21-00836-f001:**
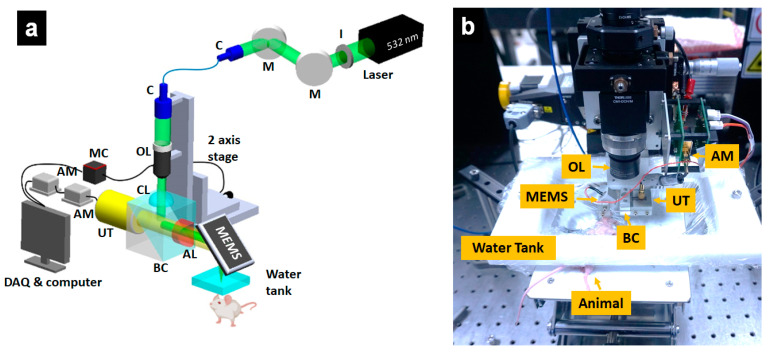
Optical-resolution photoacoustic microscopy (OR-PAM) system to monitor the peripheral vasculature of the mouse ear. (**a**) Schematic of OR-PAM, (**b**) Photograph of the OR-PAM probe part. I, iris; M, mirror; C, collimator; OL, objective lens, CL, correction lens, TR, transducer; BC, beam combiner; AL, acoustic lens; UT, ultrasound transducer; AM, amplifier; MC, motion controller; MEMS, MEMS scanner.

**Figure 2 sensors-21-00836-f002:**
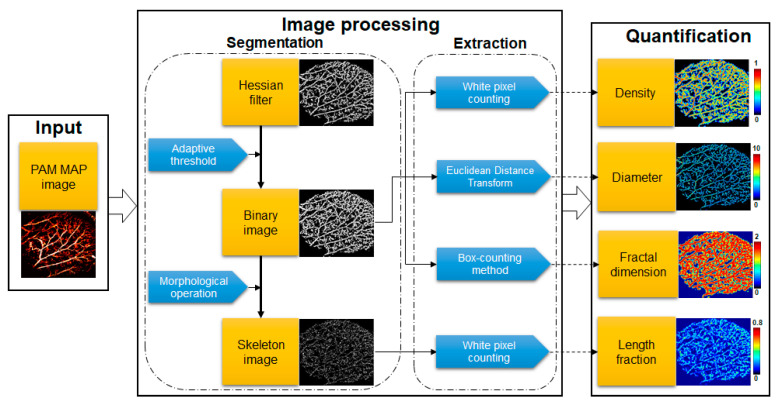
Flow chart of the quantitative OR-PAM image evaluation process.

**Figure 3 sensors-21-00836-f003:**
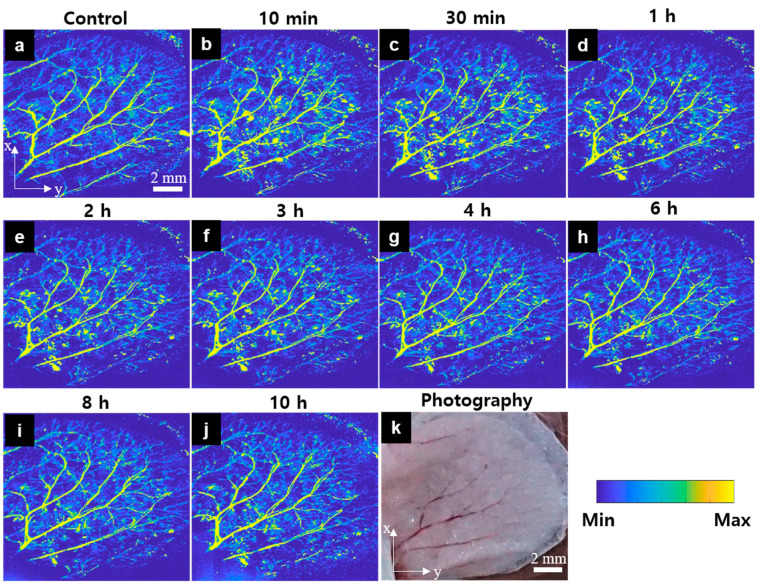
In vivo OR-PAM MAP images of the peripheral vasculatures of a mouse ear after carfilzomib solution injection (Movie 1). (**a**) OR-PAM MAP image before carfilzomib injection, (**b**–**j**) OR-PAM MAP images after carfilzomib solution injection during 10 h of observation, (**k**) Photograph of the corresponding mouse ear.

**Figure 4 sensors-21-00836-f004:**
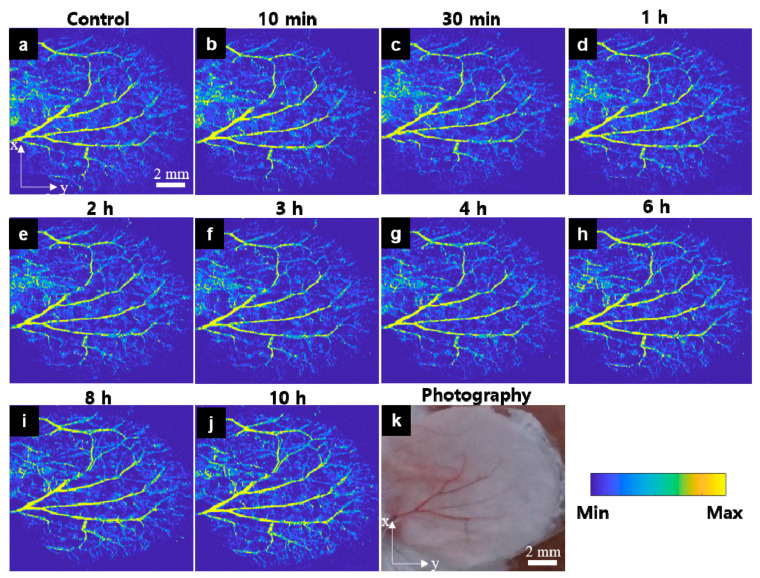
In vivo OR-PAM MAP images for the peripheral vasculatures of the mouse ear after bortezomib solution injection (Movie 2). (**a**) OR-PAM MAP image before bortezomib injection, (**b**–**j**) OR-PAM MAP images after bortezomib solution injection within 10 h, and (**k**) Photograph of the corresponding mouse ear.

**Figure 5 sensors-21-00836-f005:**
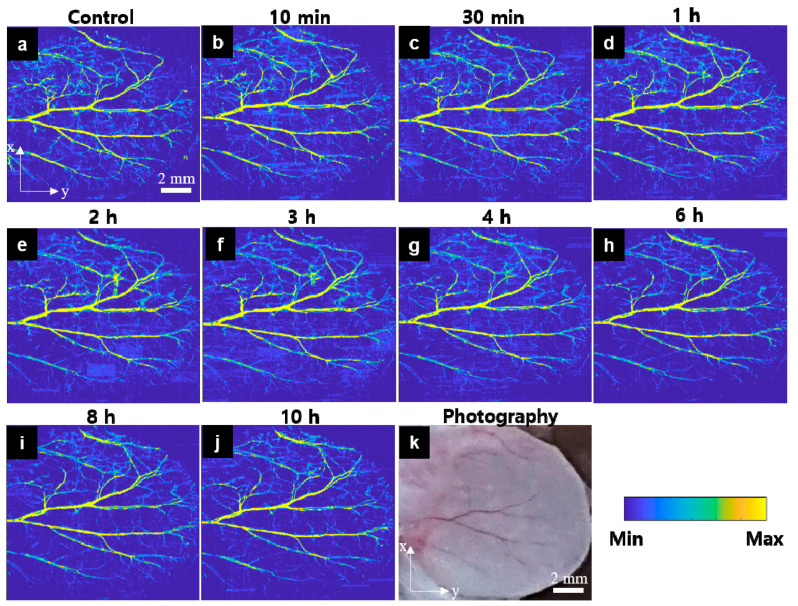
In vivo OR-PAM MAP images for the peripheral vasculatures of the mouse ear after PBS injection (Movie 3). (**a**) OR-PAM MAP image before PBS, (**b**–**j**) OR-PAM MAP images after PBS solution injection within 10 h, and (**k**) Photograph of the corresponding mouse ear.

**Figure 6 sensors-21-00836-f006:**
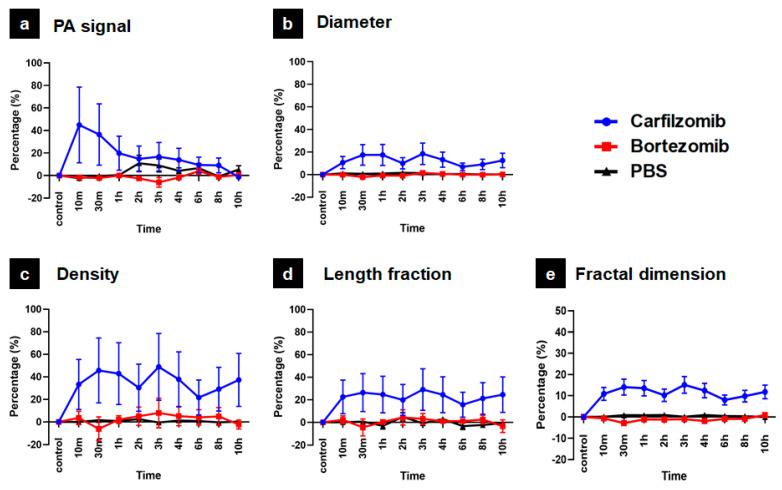
Quantitative evaluation results among carfilzomib, bortezomib, and PBS. (**a**) PA signal, (**b**) Diameter, (**c**) Density, (**d**) Fraction length, (**e**) Fractal dimension.

**Table 1 sensors-21-00836-t001:** The formulas of quantitative parameters.

Parameters	Formula	
Diameter	$\mathrm{Diameter}=\frac{\sum_{i=1}^{m}\sum_{j=1}^{n}E\left(i,j\right)}{S\left(i,j\right)}\;\left(1\right)$	E(i,j): Euclidean distance transform S(i,j): White pixel of skeleton map
Density	$\mathrm{Density}=\frac{\sum_{i=1}^{m}\sum_{j=1}^{n}B\left(i,j\right)}{T\left(i,j\right)}\;\left(2\right)$	B(i,j): White pixel of the binary map T(i,j): Total pixels appeared on the binary map
Length fraction	$\mathrm{Length}\;\mathrm{fraction}=\frac{\sum_{i=1}^{m}\sum_{j=1}^{n}S\left(i,j\right)}{T\left(i,j\right)}\;\left(3\right)$	S(i,j): White pixel of skeleton map T(i,j): Total pixels of skeleton map
Fractal dimension	$\mathrm{Fractal}\;\mathrm{dimension}=\frac{\log(N_{r})}{\log(\frac{1}{r})}\;\left(4\right)$	r: Size of the unit box $N_{r}:$ Number of boxes
PA signal	$\mathrm{PA}\;\mathrm{signal}=\frac{\sum_{i=1}^{m}\sum_{j=1}^{n}I\left(i,j\right)}{T\left(i,j\right)}\;\left(5\right)$	I(i,j): Intensity at point (i,j) of the MAP image T(i,j): Total pixels of the MAP image

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
