# Peer review of "Pilot Study: Quantitative Photoacoustic Evaluation of Peripheral Vascular Dynamics Induced by Carfilzomib In Vivo"

_sensors, 2021, doi:10.3390/s21030836_

Round 1
Reviewer 1 Report
This work shows an intersting method utilizing PAM to investigate the adverse effects on Carfilzomib on peripheral vascular dynamic. Five key related parameters were retrived from the PAM image sequence over 10 hrs and the trends have successful indiate the vascular dysfunction from the injection of Carfilzomib.
The selected parameters, in two aspects, i.e. morphologically(diamter, density, length fraction, fractal dimension) and funtionally(PA signal), show significant postinjection variation compared from the control group(PSB). Among them, in the functional pespective, the author should put further justification on the change in the PA signal. Other literatures show that the change in the PA may attribute to several factors, including the SO2 level, temperature etc. The clear justification on the mechanism can help to retrive more functional infomations and add more value of this method in speaking of physiological significance.
Author Response
Thank you for your comment.
We prepare the response for your comment.

Reviewer 2 Report
The authors studied the effect of proteasome inhibitors on peripheral vasculature using photoacoustic microscope (PAM). Major claims fo the article is the following:
1) Studied dynamics of peripheral vasculature under the effect of carfilzomib for 10 hrs.
2) Comparison with bortezomib and PBS.
3) Performed quantitative vascular evaluation process to study the difference.
This is an interesting work on quantitative study on the effect of proteasome inhibitors showing the potential of PA imaging. To improve the quality of the work I have the following suggestions.
1) In the abstract state what difference you observed between the three drugs based on the PAM study.
2) Authors should motivate the use of metrics: diameter, density, length factor, fractional dimension and PA signal strength. Why these metrics are important for the specific application? Please define the parameters/metrics for the general audience and motivate the computation shown in Fig.2 and Table 1.
3) Please mention the data acquisition time to image the whole ear and how big is the imaged area?
4) There is some movement evident in the images. Does this introduce artifacts in the image and how does it affect the analysis?
5) Major drawback of this article is that there is no conclusion to the whole study. PAM and quantitative vascular evaluation process can be used in peripheral vascular monitoring is not a conclusion for this study. Using PAM for vascular imaging is well known. Clearly stating the effect of the three drugs can be a solid conclusion.
6) The error bar in Fig 6 is quite big. Is the difference in the metrics used in Fig. 2 between carfilzomib, bortezomib and PBS statistically significant?
7) How many animals per treatment method were used (not mentioned). If it is just one animal how can the authors claim the difference at all?
8) Authors mentioned that “PA signal is not a vasculature property”, what not? Please explain the reason for the initial increase and then decrease of PA signal in carfilzomib case and an increase around 2 - 6 hrs for PBS injection.
Author Response
Thank you for your fruitful comments.
We prepare the responses for your comments.

Round 2
Reviewer 2 Report
The authors addressed all the comments properly.